# Lifestyle Properties, Ecosystem Services, and Biodiversity Protection in Peri-Urban Aotearoa–New Zealand: A Case Study from Peri-Urban Palmerston North

**Diane Pearson** 🆔

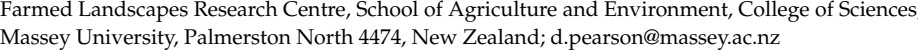

Farmed Landscapes Research Centre, School of Agriculture and Environment, College of Sciences, Massey University, Palmerston North 4474, New Zealand; d.pearson@massey.ac.nz

**Abstract:** Intensive agriculture and urbanization are putting pressure on natural capital in Aotearoa–New Zealand (NZ), with native ecosystems and water quality suffering degradation. As the population has increased, so development has pushed into the rural–urban fringe. Over the last 30 years, the number of lifestyle properties in NZ has increased dramatically. Many of these properties have been developed on some of NZ's most productive soils, meaning a loss of provisioning services from this land. However, given their location, these developments present new opportunities for the enhancement and protection of other ecosystem services. This paper presents the findings of an exploratory study conducted on lifestyle block residents in peri-urban Palmerston North. The results showed that these residents have a good sense of environmental stewardship and a desire to plant native species, improve connectivity, and protect their land from the invasion of pests and weeds. These residents are also quite community-focused and protective of their special place. This creates an excellent basis from which to encourage greater collaborative action towards protecting and enhancing biodiversity and to put in place land management strategies that can enhance natural capital and assist in other ecosystem service protection serving to improve the landscape ecology of peri-urban environments.

**Keywords:** landscape ecology; natural capital; urbanization; environmental stewardship; sustainable landscape management; environmental challenges

## 1. Introduction

Aotearoa–New Zealand (NZ) is internationally renowned for its natural environments and rural landscapes. Of the country's 26.8 million ha of land area, only about 0.9% is considered urban [1], and the countries rural-based industries play a significant part in the economy. Agriculture is NZ's largest industry, and it is the world's largest exporter of dairy, whilst tourism ranks second to dairy farming in terms of foreign exchange earnings [2–4]. Yet, its population is largely urban. In 2020, there was 86.7% of the population living in cities and urban areas, and this number is rising [5]. This means that less than 20% of the population currently live rurally, and a growing urban-rural divide amongst the population is often noted [6].

Ecosystem services were defined by Costanza et al. [7] as "ecological characteristics, functions, or processes that directly or indirectly contribute to human well-being, that is, the benefits that people derive from functioning ecosystems" with the ecosystems that provide the services being often referred to as "natural capital" [8]. From an environmental frame of reference, an increase in urban development, combined with more intensive agricultural practices, is putting pressure on NZ's natural capital with native ecosystems and water quality suffering degradation [9–11]. This degradation is cause for heightened concern in NZ, with considerable pressure for land managers and developers to operate more sustainably [6]. In terms of the total land area of NZ, only about 32% is currently protected in reserves and national parks [12], and between 2012 and 2018, indigenous

land cover in NZ was noted to have decreased by 12,869 ha [13], further highlighting the fact that natural capital is under threat. This means that there is increasing demand for more "off reserve" conservation and a need for private landowners to play their part in protecting and enhancing important ecosystem services.

Since NZ landscapes form part of the national and international NZ identity, New Zealanders place a high value on both their "farmscapes" and picturesque natural landscapes. Māori (the first inhabitants of NZ) have a strong connection to landscapes and nature. This means that loss or damage to important ecosystems and the services they provide can have a detrimental impact not only on natural capital but also cultural capital. Mātauranga (Māori knowledge, which incorporates traditions, values, concepts, and world views) and national well-being can be degraded through damage to endemic ecosystems. Therefore, an important consideration for sustainable landscape planning needs to be the cultural ecosystem services that NZ landscapes provide, as well as the value these landscapes play from provisioning, supporting, and regulatory perspectives.

In recent years, the focus of government environmental policy in NZ has been directed towards farmers and farm practices to attain higher environmental sustainability goals. The policy has been targeted especially towards trying to achieve a reduction of nitrate and methane emissions from agricultural production (e.g., see National Policy Statement for Freshwater Management 2020 [14] and Climate Change Response (Zero-Carbon) Amendment Act 2019 [15]). There has also been increasing attention on enhancing biodiversity and creating multifunctional landscapes for the multitude of ecosystem services benefits these approaches can provide [16,17]. Emphasis has also been placed on how to design future landscapes through incorporating the value of "land sharing" and diversification of farmland for ecosystem service enhancement and protection [16,18]. The application of regenerative farming practices in NZ has also recently encouraged research as well as practical attention to help ensure provisioning services can be perpetuated whilst protecting important natural and cultural ecosystem services [19,20]. Getting the balance right between preserving and enhancing ecosystem services is vital to maintaining the sustainable landscapes, lifestyles, and livelihoods that New Zealanders want. This means creating a highly productive and sustainable agricultural sector in a backdrop of natural landscapes that can be enjoyed by locals and international visitors alike.

However, agricultural environmental issues are not the only challenges for this "green dream". Urbanization presents considerable problems too. The population of NZ is predicted to increase by an average annual rate of about 0.8%, taking the NZ population to over 5.1 million by 2031 [21]. It is expected that the North Island will see more growth than the South Island, so some regions will experience more pressure for development than others, heightening the stress on natural capital and ecosystem services in these areas [22]. In the last couple of years, there has also been an increased demand for housing, which may have been compounded by many New Zealanders who would normally go overseas not leaving NZ due to the COVID-19 pandemic [23]. This is putting even greater strain on regional urban centers causing house prices to rise to an all-time high, such that many people are now potentially at risk of being excluded from house ownership if housing stocks are not adjusted accordingly [24]. As a result, urban developers must push their developments increasingly on to "greenfield" sites so that supply can meet the demand. In a quest to address housing shortages, and as urbanization strategies roll out, residential and commercial development will likely increasingly encroach on to current peri-urban and rural land.

Peri-urban areas are seen as being transition zones between urban and rural zones [25,26]. They are often made up of mixed land use and can be subject to rapid land-use change [25]. They can also be made up of a mixture of residents ranging from newcomers to long-established residents meaning that the socio-economic make-up is variable. From an agricultural production perspective, the loss of high-grade farmland that is currently found close to urban centers presents issues for sustainable resource management [27]. Good quality agricultural land is being taken at an increasing rate to create residential areas,

locking up land permanently [27]. The loss of the important provisioning services provided by high-quality soils presents a considerable challenge for NZ and is one that needs to be carefully considered in both urban and economic planning [27].

Amid this change, more people are searching for an idyllic lifestyle within a commuter belt associated with the "tree change" phenomena [28], which has also led to a trend in farmland being subdivided into the smaller semi-rural sections that are known as lifestyle blocks [29].

Land Information New Zealand [30] (p. 60) defines a lifestyle block as being "generally, in a rural area, where the predominant use is for a residence and, if vacant, there is a right to build a dwelling. The land can be of variable size but must be larger than an ordinary residential allotment. The principal use of the land is non-economic in the traditional farming sense, and the value exceeds the value of comparable farmland". The definition of a lifestyle block implies that most residents do not rely on their land for a primary source of income through agricultural or horticultural use. With increasing numbers of lifestyle blocks in NZ, there is not only a recorded loss of important versatile soils but also potential for environmental damage, especially if block owners do not have adequate knowledge and understanding of appropriate sustainable land management and best practice natural resource management practices when managing their land.

In 2011, 175,000 lifestyle blocks occupied 873,000 ha of land across NZ [27]. Over 40% of these had been developed since 1998, meaning growth of lifestyle blocks was averaging 5800 new properties per year [31]. By 2018, this rate had slowed a bit with just over 186,000 lifestyle properties covering nearly 900,000 ha [32], but subdivision for these properties continues.

Despite lifestyle block properties occupying a significant amount of NZ land stock, research on this type of property has been relatively limited to date. Of relevance to this paper is a 2004 nationwide survey of residents that recognized that assessment of environmental impacts on these blocks was informed only by resource consent applications and impact assessments made by regional councils [33]. This results in relatively little being known about land-use practices and even less being known about the behavior of residents [33]. Understanding the motivations and drivers that determine environmental attitudes and behavior on lifestyle blocks, however, is a worthy subject for research, especially as environmental issues in NZ become more significant. There is also merit in better understanding the thoughts and practices of this group of residents as they grow in number and can play a significant management role in the urban–rural fringe. Being able to determine useful approaches to encourage pro-environmental behavior and sustainable practices amongst lifestyle property residents could prove very beneficial both from a rural and urban planning and environmental management perspective.

This paper takes a landscape ecological perspective to explore what private landowners on lifestyle blocks can do towards preserving and enhancing natural capital and ecosystems services in peri-urban environments. The aims of the study were to determine the sense of environmental stewardship that lifestyle block residents have and what forces act as barriers and motivators towards environmental management of their properties. In doing this, it was hoped to address the research question of whether lifestyle block residents could provide the collective capacity to improve and enhance the landscape ecology of peri-urban landscapes. Although reserve networks and organizations such as the Department of Conservation play an important role in protecting ecosystems in NZ, the ability to expand ecosystem protection across greater areas of land (i.e., on collective small blocks) to increase habitat connectivity and promote biodiversity on land outside of the reserve network through community participation, could add exceptional value to the protection of NZ's natural capital.

While NZ farmers are increasingly expected to improve their environmental management standards to meet policy requirements, little is known about the land management and environmental practices being undertaken on lifestyle blocks. Therefore, the study presented in this paper attempted to be a first step in addressing this knowledge gap in

NZ. It is hypothesized that knowing more about the environmental actions and the attitudes, motivators, and barriers to environmental management, as well as determining how lifestyle landowners could be encouraged to engage more in environmental practices, has the potential to increase ecosystem services provision and reduce adverse environmental impacts due to poor land management practices.

This paper summarises the findings of an exploratory study conducted on the Turitea Valley, in the urban–rural fringe of Palmerston North *Te Papa-i-Oea* in the Manawatū region of the North Island, NZ (see Figure 1). Like many regions in NZ, the Manawatū region faces considerable environmental challenges that have arisen due to land clearing and the intensification of agricultural practices. Pollution reduction, conservation of biodiversity, minimization of land degradation, and erosion are the serious focus of environmental concern for the regional and city councils. The environmental damage that can result from intensive agriculture is the focus of research and community action, especially around water quality and contaminated "runoff", which is polluting the region's water systems, especially the Manawatū River that flows through Palmerston North [33–35]. Intensive urban development in the region also has potential for environmental implications. Expected population growth linked to regional development planning and a growing number of residents opting for a "tree change" will put increasing pressure on the urban–rural fringe. It is hoped that this study will help to inform thinking that can assist practical environmental management in the peri-urban area. It is also hoped that it will encourage more in-depth landscape ecological research into how lifestyle block residents can contribute to wider catchment-scale environmental management, not only in NZ but also overseas.

## 2. Materials and Methods

### 2.1. Study Area

Turitea is a peri-urban suburb of Palmerston North. Palmerston North is located at a latitude of 40.3545° S and longitude of 175.6097° E. It lies 142 km north of NZ's capital—Wellington. As of June 2021, the population of Palmerston North city was 81,500 people, with 5770 people located in surrounding settlements and peri-urban areas [36]. Turitea is located on the southern side of the Manawatū River, spanning from the city boundary to the Tararua Ranges and further south towards the small town of Tokomaru (see Figure 1). The catchment in which it sits provides about 60% of the water for Palmerston North, and the hills at the top of the valley are to be occupied by what will be the largest wind farm in NZ with a total capacity of 222 MW. Construction on the wind farm started in 2019, and when finished, it will have the ability to meet the needs of 118,000 average homes [37]. Farmland in the form of hill country surrounds the suburb on its rural side. The land use within the catchment demonstrates the important provisioning services that the location contributes towards both the city of Palmerston North and NZ.

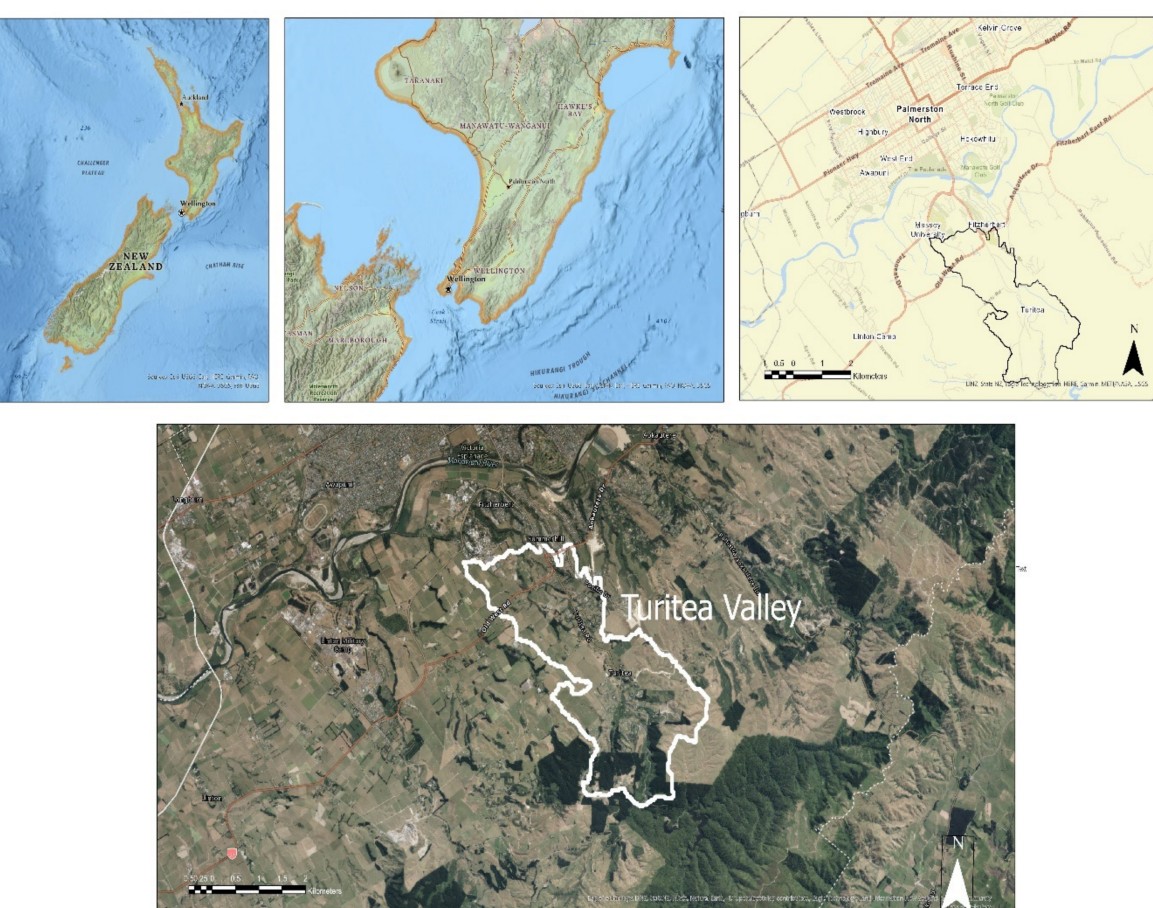

**Figure 1.** Map showing Palmerston North and the Turitea Valley study area.

From an important environmental perspective, the Turitea Reserve is also found within the area of interest. This covers about 4000 ha, making it the largest reserve accounted for by the Palmerston North City Council [38]. The reserve is significant for biodiversity and is important for recreation. Risks and challenges to the reserve have been noted as protecting the important water supply for Palmerston North from contamination, managing highly significant native flora and fauna and ecosystems at risk from introduced species and fire, managing important amenity value and recreation opportunities as well as some important cultural and historic sites for the local area, whilst also managing the reserves interaction with electricity generation and the harvest of some neighboring radiata pine (*Pinus radiata*) plantations [38]. The area is currently being managed as a predator control area and has undergone restoration of native plants. There has been a recent translocation of the Toutouwai, North Island Robin (*Petroica longipes*). They are endemic to NZ and are classified as at 'risk/declining' and are locally extinct in most of their range [39]. As the population of the Toutouwai grows, it is hoped they will spread down the valley towards the city [40], but to do this, they will need habitat connectivity.

This study focused on a subset of the suburb known as the Turitea Valley. Note this does not cover the entire landscape unit of Turitea as defined by the Palmerston North City Council. The Turitea Valley lifestyle blocks assessed in this study are located below the reserve in the catchment through which the Turitea stream runs. Many of the houses along the main roads in the study area were built in the 1990s and early 2000s [41]. Habitat restoration has taken place on council-acquired land along the riparian strip next to the Turitea stream under the "Green Corridors" project [42]. This offers potential for increasing the size and connectivity of habitats on privately owned land that adjoin the "green corridor". Extending habitat protection and restoration into the catchment

requires co-operation and contribution from owners of lifestyle blocks in this area. The best approach to achieve this is through engaging landowners in conservation efforts. With the Turitea Stream located in this area, environmental management on these properties is also important for water quality outcomes. This means that the Turitea Valley has several economic, social, cultural, and natural values.

As little demographic research has previously been undertaken within the Turitea Valley area as opposed to the Turitea suburb, there are limited data available regarding the demographics of these specific residents. The entire suburb of Turitea, according to the 2013 census, had a population of 1878 people across 639 occupied residences [43]. In terms of age demographics at the 2013 census, 23.3% of residents were aged under 15; 67.4% were aged between 15–64, and 9.3% were aged over 65. With almost one-quarter of residents under 15, it is evident that a large proportion of residents are families; indeed, 57% of households were classed as either single parents or couples with children [43]. The professions of those employed (in order of most common) were retail trade (23.8%); manufacturing (18.8%); agriculture, forestry, and fishing (9.4%); healthcare/social assistance (8.8%); and professional/scientific/technical services (8.1%) [43]. Sixty-nine percent of the working adult population drove to work with a private car, truck, or van, highlighting the fact that many residents commute to work. Just over 80% of the population were owner-occupiers. In terms of ethnicity, 91.5% identified as European; 11.4% Māori; 1.2% Pacific peoples; 2.4% Asian; 0.7% Middle Eastern, Latin American, and African; and 3.0% other ethnicities. Those aged fifteen years and over holding a formal qualification made up 86.8% of the population, with 31.2% having a bachelor's degree or higher as their highest qualification. These proportions amongst Turitea residents differed significantly at the time from those of the entirety of Palmerston North, suggesting the demographics of residents in this area is different from that of the urban population [43].

Development in the wider Palmerston North area has followed a steady pattern of growth, and for the last 20 years, housing development has mostly been 'greenfield' with some infill subdivision and the creation of lifestyle properties [44]. Rural and rural-residential growth made up 12% of demand for residential growth in Palmerston North City [45], and in 2013, a quarter of new rural residential development took place in Turitea [46]. In 2018, it was estimated that the area of available residential land in the Palmerston North area exceeded demand by 20% and that there was significant capacity for new lifestyle blocks [47]. Pressure on Turitea comes from an increase in demand for "greenfield" housing development associated with the expansion of the nearby "urban" suburb of Summerhill/Fitzherbert and greater subdivision of the neighboring peri-urban suburban of Aokautere [44]. Any subdivision in these areas currently requires resource consent approval under the Resource Management Act 1991 from Palmerston North City Council [48].

As the threat to subdivide within the peri-urban environment to create higher density smaller residential units becomes greater, it is increasingly important to recognize the valuable role that peri-urban environments can play in protecting natural capital so appropriate planning control measures can be put in place. It is also vital to maintain as much native vegetation and habitat in the peri-urban fringe as possible to maintain ecosystem services.

*2.2. Methodology*

This study was conducted using qualitative research with targeted lifestyle block residents on blocks between 0.4 and 50 ha in size in the Turitea Valley. Data were gathered via postal invite to an online survey. The survey was carried out using the Qualtrics platform. Since an invitation to participate was sent via the postal service, this limited the survey to those properties that receive postal deliveries to their on-street post box. From examining Google Maps covering the main streets within the catchment, it was determined that there were between 145 and 155 eligible lifestyle properties in this area. Deliverable rural addresses for these properties were found using the NZ Post website to manually

enter each house location. This process identified 145 properties, and these were used to solicit participation.

The letter of invite informed the residents of the study and its aims. Residents were also informed of the end date for responding to the survey, as well as given the online link to the survey and an estimation of how long it would take to complete, i.e., 15 min. All invited participants were made aware that they had the right to decline to answer any question, to withdraw from the survey and not submit a response, to ask any questions about the survey at any time, to provide information on the understanding that responses were anonymous. This study was developed following the Massey University Code of Ethical Conduct for Research, Teaching, and Evaluations Involving Human Participants (2017), and the project was peer-reviewed and deemed low risk, which meant that it was only peer-reviewed and did not need more formal Human Ethics approval.

Before circulation, the survey was piloted using four uninvolved (non-biased) individuals to ensure it was easy to understand, had a logical layout, and avoided unnecessary complexity for non-scientific audiences. Feedback received from the pilot was taken into account, and minor changes were made to remove ambiguity and ensure clarity, given the survey would be completed online. The survey was tested for useability on both a computer and a mobile phone device to ensure it was compatible with different screen sizes.

The information sheet that was supplied specified that the survey targeted residents within a particular property range (i.e., between 0.4 and 50 ha), which prevented ineligible properties from taking part. Although this introduces purposive sampling by selecting only candidates that fit certain criteria, it ensured that all results would be relevant to the study rather than receiving a higher response rate but having to eliminate unusable responses. Given these factors, it should be noted that the sample selected to participate is deemed a convenience sample.

Letters were sent out on the 20 December 2018. Originally the online survey was scheduled to close on the 12 January 2019. However, limited responses were received by this date, potentially due to the overlap with the NZ summer/Christmas holidays, which may have affected the response rate. A follow-up letter was posted on the 16 January 2019, extending the survey and giving residents until the 18 February 2019 to reply.

The survey was divided into five sections: (i) Occupier demographics: general questions about the length of residency, age, education, and income. (ii) Property and land use information: questions about the size of the property, land use, natural features, and motivations. (iii) Environmental awareness, perceptions, and concerns: several Likert scale questions about environmental issues as well as motivations, barriers, and practices. (iv) Knowledge and information: sources of information and current/potential support platforms. (v) Life in the Turitea: open-ended questions about what residents thought about the Turitea Valley.

The survey was largely analyzed within the Qualtics platform with summary statistics generated. Open-ended questions were evaluated and reviewed using manual processes associated with content analysis to create themes of responses.

## 3. Results

A total of 32 responses were received, giving a response rate of 22%. No responses were required to be eliminated as all met the criteria. A response rate of 22% is considered adequate for this exploratory study. It equates to a confidence level of about 80% and a 10% margin of error. It is acknowledged that some bias may be present in the survey method because participation was voluntary, and therefore people might have been more likely to participate if they had concerns about the environment and land management. A relatively high non-response rate might also contribute to bias. There were cases where respondents did not address a particular question. Although respondents mostly answered all questions, they could refuse to answer any question they deemed inappropriate.

### 3.1. Respondent Demographics

The demographic characteristics of the respondents can be seen in Figure 2, with results presented as percentages. The survey showed that when respondents were asked which category of property owner they identified with, the most common response was "lifestyler". In terms of ownership, 97% of respondents owned or were buying the property they lived on, and more than half of respondents had lived on their property for more than ten years. Just 30% had lived on their property for two years or less. Only a small proportion of respondents received an income from their property, and none of these were primary incomes.

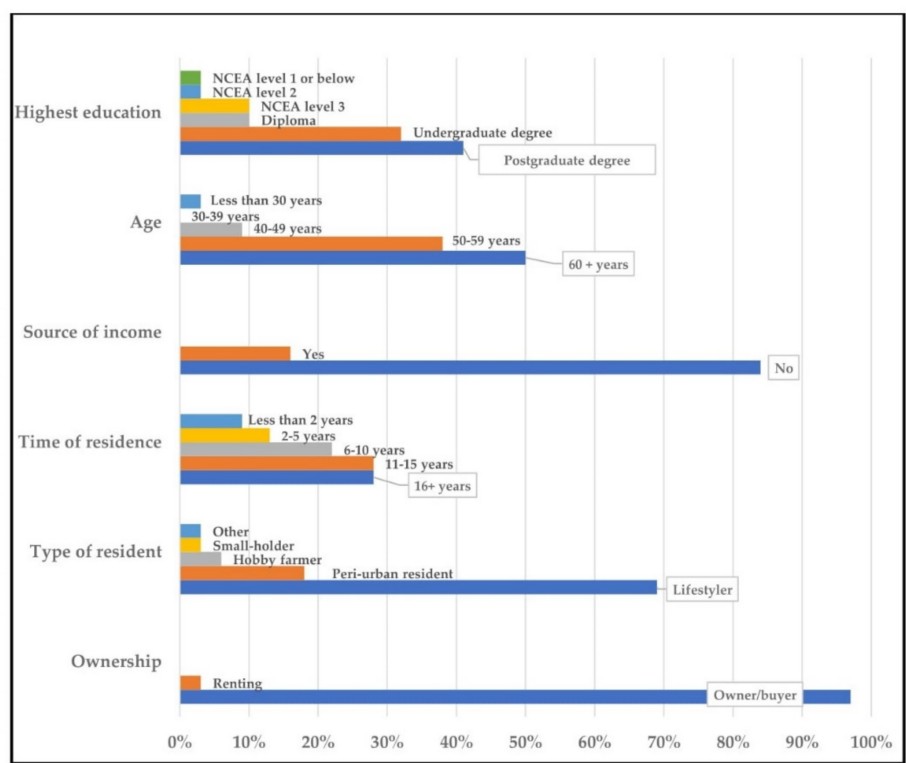

**Figure 2.** Demographic characteristics of the respondents by percentage.

Respondents were asked to state their occupation; these were then classified according to categories used in Stats NZ census data. The most common occupation was retired/pensioner (42.31%), with professionals making up the next highest category (38.46%). Community and personal service workers made up 11.54% and technicians and trade workers 7.69%. The age of respondents is consistent with the employment status, i.e., half the respondents were over 60 years of age, and only about 3% were less than 30 years. Most respondents had completed tertiary-level qualifications.

### 3.2. Property Details and Information Relevant to Land Use and Ecosystem Service Provision

Factors that contributed to the respondents' motivation for owning/living on a lifestyle property were also surveyed. Results showed that deriving an income did not influence respondents' motivation, but that space, privacy, peace, quiet, and a relaxing and healthy environment were the main motivational factors; also important was the fact that living on a lifestyle property in the peri-urban area was deemed to be less pressure than urban living. Other important factors included great views, clean air, the ability to have animals and land for recreation, and the desire to produce own food and be more self-sufficient.

The most common property size category was 0.4–2 ha. These properties made up just over half of the properties surveyed. The main land use activities carried out on

properties can be seen by percentage occurrence in Figure 3. Of note is that household gardens/recreation, vegetable gardens, homes for pets, and use for livestock and breeding are the most common land use activities on lifestyle properties, while 30% of respondents engaged in use for wildlife protection and conservation.

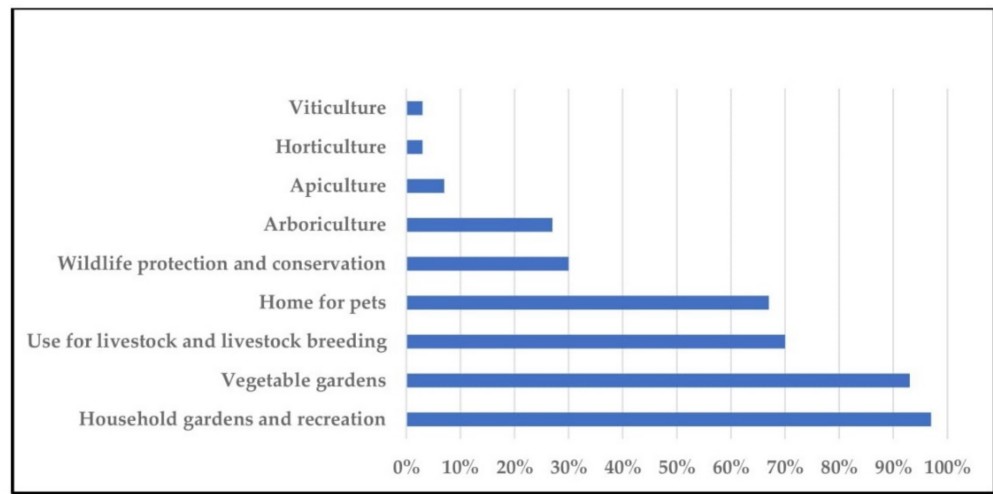

**Figure 3.** Percentage of properties engaging in different land use activities.

Native trees were the most common natural feature on properties, with 70% of respondents' properties containing native trees and 53% of respondents having native shrubs/grassland on their property. Natural ponds/wetlands were present on 20% of properties, with streams present on 27% of properties and 13% of respondents having access to groundwater.

One of the survey questions asked whether owners had noticed an increase in native birds after implementing environmental practices or changes in land management to their property; 70% reported having seen an increase in native birds. Changes that were undertaken by those who had seen an increase in native birds included planting native trees/shrubs, replacing pines with natives, trapping predator species, and providing bird feeders.

When asked about non-native pest species, 85% of respondents said weeds were a nuisance on their property, while animal pests were a problem on 81% of properties. Weed species that were identified causing an issue included thistles, foxgloves, mallow, barley grass, gorse, blackberry, passionfruit vine, elderberry, buttercups, dock, prairie grass, convolvulus, agapanthus, wild ginger, and tradescantia. Animal pests observed on properties included rats, stoats, weasels, possums, sparrows, starlings, cats, mice, rosellas, and magpies.

When asked about aspirations for future land use, 57% of respondents wanted to keep their property as it was, 11% wanted to use it to generate more income, and 18% wanted to use it to become more self-sufficient. Other responses that were listed from an ecological perspective included replacing pines with natives, building an eco-house, developing native bush and removal of pest species, and increasing the size of their orchard.

### 3.3. Environmental Awareness, Perceptions, and Concerns

Respondents were asked to rate their level of environmental awareness and to answer questions about their perspectives on habitat connectivity and environmental stewardship. Most of the respondents felt that they had a high level of environmental awareness, with 58% ranking themselves as having "high awareness", 11% ranking their awareness as "exceptionally high", and the remaining 31% ranking their awareness as "moderate". None of the respondents indicated they had low environmental awareness. In terms of their sense of environmental stewardship and responsibility for their block, over 62% rated

themselves as having "a great deal" or "a lot", whilst another 31% rated themselves as having "a moderate amount".

The survey showed that the most common land management activities undertaken on their properties to improve it environmentally was to apply manure, aimed at improving the soil, with 88% of respondents having done this. Engaging in practices to minimize erosion was undertaken by 54% of respondents; fencing water bodies was done by 15% of respondents while receiving advice on chemical use and storage had been done by just 7% of respondents. When asked what specific practices were undertaken out of consideration for the environment, results showed that all respondents have sprayed or pulled weeds from their property, with over half of all respondents having: created habitat for native birds (71%), trapped/poisoned pest animals (68%), had energy-efficient heating installed (75%), had insulation installed (68%), and managed stock numbers (54%).

Figure 4 shows how important maintaining habitat connectivity was to respondents personally; three-quarters of respondents ranked this as being very important or moderately important, whilst 20% deemed it to be somewhat important, and only one person deemed it not to be important at all.

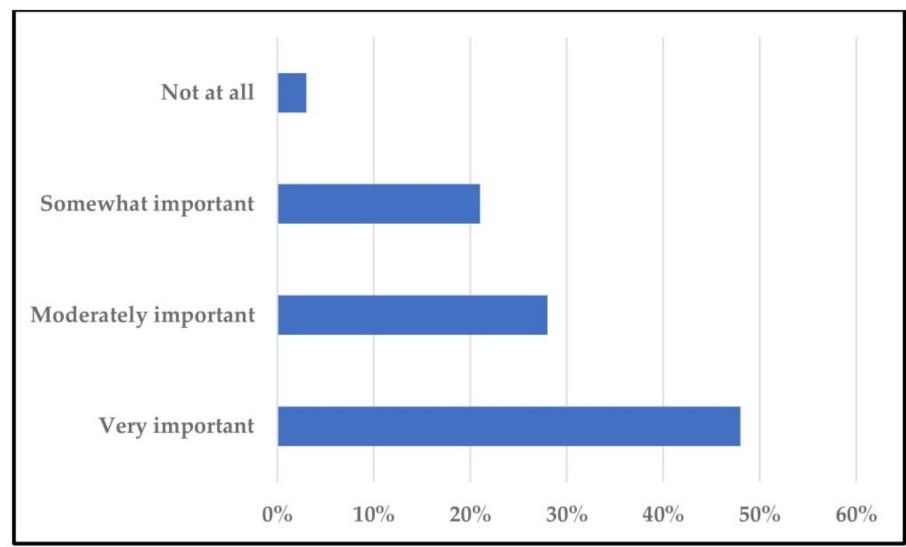

**Figure 4.** The importance of habitat connectivity to lifestyle block residents by percentage response.

Respondents were asked whether they thought their lifestyle choices were having a negative environmental impact. Just over half of respondents (58%) indicated they did not think so, with 31% saying "yes" and 3% saying that they "did not know". On the whole, the lifestyle choices recorded did support reducing environmental impact, with 96% of respondents engaging "all the time" in some form of recycling, 78% engaging in composting "all the time", and another 7% doing it "very frequently", 34% minimized electricity consumption "all the time", and another 38% did it "very frequently", and 20% reduced personal consumption to reduce waste "all the time" with another 55% doing it "very frequently".

Respondents were also asked to indicate whether they thought that any potential environmental threats could come from their property. Almost half of the respondents believed their properties did not pose any threat to the environment, but of the threats identified, 14% thought the spread of pests/weeds might be a problem, 10% thought water pollution could be a potential issue, 7% thought soil erosion, and another 7% indicated that removal of native vegetation/biodiversity loss might be having an impact. The "other" category made up 10% of responses, and this included "felling pines", "energy from commuting", "waste disposal", and "emissions from stock grazing". When asked if they would be willing to change their land management practices if their land management activity was more damaging than originally thought, 44% of respondents were willing to

change their practices; 44% were willing only if it was not too difficult or expensive, while 11% indicated that they were satisfied with their land-use practices as they were.

The top three environmental concerns chosen from a given list showed that respondents deemed climate change, freshwater/river quality issues, and waste management as the most pressing issues for New Zealand. Agricultural emissions, urban sprawl, land clearing, and invasive pests were in the next band of concerns. Interestingly, loss of biodiversity was ranked lower than loss of productive soil. Answers included under "other" were "global overpopulation", "human population growth", and "use of pesticides and hormones".

Figure 5 shows the most common concern environmentally for the Turitea Valley was deemed to be the spread of pests and weeds; 37% of respondents thought this was a threat. Other important threats were water pollution, soil erosion, and the removal of native vegetation, and the loss of biodiversity. Threats identified in the "other" category included "forestry operations", "wind farm development", "traffic and housing development". These were obvious concerns given the development going on in the area at the time, i.e., harvesting of plantations forest, the construction of the wind farm, and increasing subdivision—all of which were putting pressure on the local rural road system.

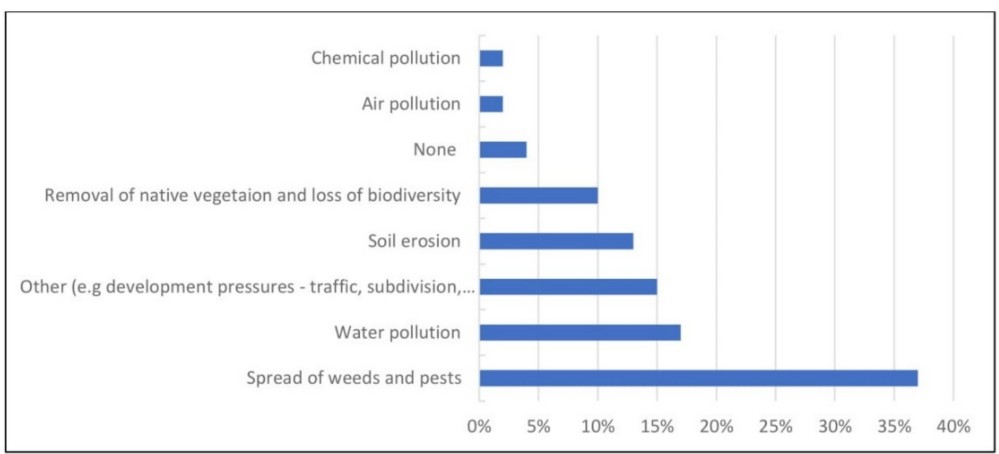

**Figure 5.** Respondents' views (by %) of the biggest threats to the Turitea Valley.

When asked if respondents felt they needed more information on particular topics, 42% of respondents felt more information was needed about reducing environmental impacts, while 35% felt there were no areas where more information was needed, and 31% felt it would be useful to get more information on land management and growing practices, and 4% thought information on better stock management practices was needed. "Sustainable lifestyle changes", "reducing food waste", "reducing animal protein", and "cost/km of owning larger cars" were responses listed under the "other" category.

In terms of the type of support platforms, respondents thought would be useful for receiving land management information; most respondents (68%) indicated a preference for online support platforms, while advisors, leaflets/books, and meetings, seminars, and forums were also deemed useful. Most respondents were not a member of a group working on environmental management within the community—59% of respondents indicated they were not involved with any such groups and "wouldn't like to be", 35% were not involved in any groups but "would like to be", and just 7% were already involved with a group working on environmental management.

When asked what motivated respondents to engage in environmental practices, all respondents indicated that they do this from personal choice; 10% of participants were also motivated by societal expectations, while 3% were also motivated by policies/regulations. When asked if they believe it was the responsibility of the individual or governments to address environmental issues, all respondents said "both".

In terms of identifying barriers to implementing changes to land management for environmental outcomes, the most common response was lack of funds, impacting on 64% of respondents, then lack of time which affected 54% of respondents. Lack of knowledge and expertise was also seen to be a limiting factor for 35% of respondents, while 4% of respondents indicated they did not want to implement any changes.

Figure 6 shows where respondents felt their knowledge was most limited and could be acting as a barrier towards them doing more. The graph shows that respondents felt that they lacked important knowledge around feral animal and pest control, as well as weed management. Soil management and erosion control, as well as good grazing practice, were also seen as areas where their knowledge could be improved to help them do more in terms of bringing about change on their property.

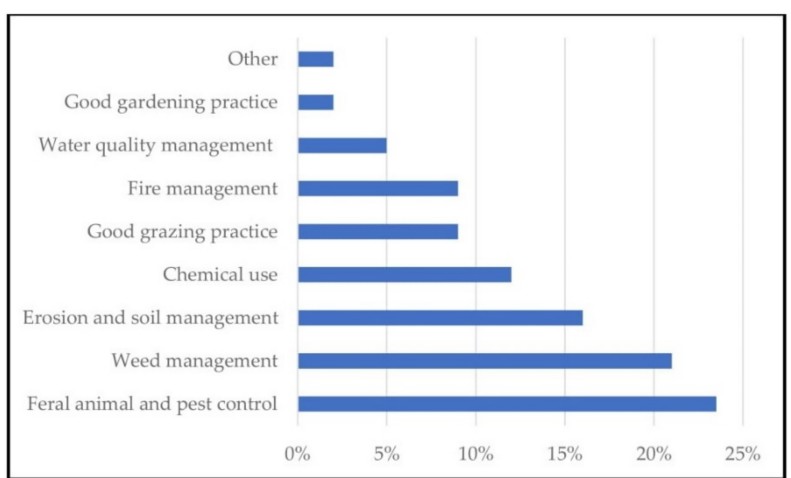

**Figure 6.** Knowledge limitations identified by respondents.

### 3.4. Life in the Turitea Valley

The last set of questions explored how people felt about the Turitea Valley. When asked whether they felt a strong cultural connection to the Turitea Valley, 93% said "yes". The survey asked respondents to state what they thought was the best thing about living in the Turitea Valley. Given the open-ended nature of this question, responses were grouped into four themes. These themes were (i) the natural landscape (chosen by 29% of respondents), (ii) rural lifestyle close to town (chosen by 40% of respondents), (iii) neighborly and friendliness of the community (chosen by 11% of respondents), and (iv) peace and quiet (chosen by 21% of respondents). Respondents were also asked what they did not want to lose from the Turitea Valley. The responses from this question were (i) rural atmosphere (with too much subdivision seen as being a significant threat to this and a recommendation made for minimum block sizes of 1 ha) (chosen by 32% of respondents), (ii) biodiversity (chosen by 29% of respondents), (iii) peace and quiet (chosen by 21% of respondents), (iv) views (chosen by 11% of respondents), and (v) neighborliness (chosen by 7% of respondents). Additional things that respondents would like to see in the area included: (i) traffic controls and road safety (chosen by 52% of respondents), (ii) environmental management—better control of weeds ad pests and water pollution management (chosen by 26% of respondents), (iii) habitat connectivity and biodiversity (chosen by 11% of respondents), and (iv) better services (chosen by 11% of respondents).

### 3.5. Summary of Survey Results

To summarize the main findings, most respondents to the survey considered themselves to be either "lifestylers" or "peri-urban residents", which is to be expected given the study took place in a peri-urban area with blocks designated "rural residential". The most common uses of the land were household gardens, vegetable gardens, livestock, and as a home for pets. Almost all the respondents indicated their motivation for living on

a lifestyle block was for peace, quiet, space, privacy, or tranquility. Just over half of the respondents believed that the biggest threat to the Turitea Valley was the spread of weeds and pests. The main limitations to engaging in environmentally friendly practices were lack of funds and lack of time, although lack of knowledge was also listed. Almost half of the respondents believed their property has no negative impacts on the environment, with many engaging in practices that helped to improve biodiversity by planting natives and reducing the spread of pests and weeds, with personal motivation being their driving factor. The top three personal environmental concerns of respondents were climate change, freshwater/river quality issues, and waste management. Locally many respondents were concerned by the state of the roads and traffic safety, as well as future subdivision affecting their rural lifestyle.

## 4. Discussion

Previous research has identified that peri-urban landscapes across the world are under pressure and can endure rapid transformation and land-use change because of urban expansion [25,26,49,50]. Complex relationships exist within these environments creating ambiguous rural and urban functions making management difficult [51]. The threat this poses to natural capital and ecosystem services emphasizes the need for better policy and planning directed towards this important fringe zone [50]. The Turitea Valley, which formed the case study for this research, is no exception in terms of ambiguity. It sits on the fringe of the expanding city of Palmerston North and, in doing so, contains and provides important ecosystem services for the city (not least acting as a vital source for the city's water supply, whilst the reserve acts as an important haven for native species and place of recreation for urban dwellers). The land clearing that created agricultural landscapes that once sat on the boundary of the city caused a loss of natural capital. Recent subdivision for peri-urban houses and infrastructure has further impacted ecosystem services by locking up good farmland, and if "lifestylers" do not manage their blocks well, other ecosystem services are at risk of experiencing further degradation. However, the potential exists to put in place measures to protect or enhance natural capital and the services they provide contributing to wider landscape environmental outcomes.

The survey conducted on a section of the resident population occupying lifestyle properties in the study site demonstrates that there is a desire for ecosystem service protection in the area. The results of this study indicate that lifestyle residents in the peri-urban area are ideal candidates to work with to protect natural capital and increase ecosystem service provision. Placing a particular focus on enhancing biodiversity and improving the landscape ecology of the area could reap the rewards due to a visible sense of environmental stewardship amongst residents. The desire for these residents to protect the lifestyle they pursue, which includes living in a peaceful environment that incorporates native species amongst household gardens and some self-sufficiency in terms of food production, also presents opportunities for environmental management. The circumstances created by their sense of community and connection to place also means greater collaborative action is a definite possibility.

In terms of ecosystem service provision, this study shows that lifestyle properties have an important role to play in supporting services. Encouraging favorable land management action amongst this population could be valuable for biodiversity protection and restoration, as well as provide expansion to important habitat for native species improving connectivity and reducing fragmentation. Most respondents in this survey valued native habitat and could help to contribute more in terms of habitat protection and restoration with some incentives.

The survey showed that most respondents felt they had good environmental awareness and encouraged biodiversity. Most respondents deemed native trees and shrubs important even though there was currently only a small amount of nature conservation formally carried out on the properties. Many respondents had planted native trees and shrubs on their properties but mostly for amenity and privacy purposes. However, since

encouraging biodiversity was deemed valuable and that anecdotally people reported that their land management practices had resulted in an increase in native birds, this indicates that without anything other than personal motivation, this group of residents is already making a difference to the biodiversity of the area. With habitat connectivity seen as important by almost three-quarters of respondents who had created habitat for native species, this could additionally indicate a willingness to expand native planting initiatives. This offers potential to expand schemes like the "Green Corridors" initiative run by Palmerston North City Council. Meaning that it is possible that the restoration that has started by planting riparian strips could be expanded onto private land to improve connectivity, reduce habitat fragmentation, and thus improve biodiversity outcomes. Other potential ecosystem services provided using increased native species planted on properties were identified by respondents as including erosion control, wastewater soakage, and windbreaks, all of which could have significant benefits for the environmental management of the area.

As well as developing more stands of native bush and replacing non-natives with natives, future land management action noted by respondents also included removal of pests and weeds. The spread of pests and weeds and the impact of these were by far the biggest threat and concerns for the area identified by respondents. As a result, all respondents had engaged in spraying or pulling weeds, and over 60% had trapped or poisoned pest species. Loss of native vegetation and biodiversity was possibly considered to be less of a threat than pests and weeds because the area had been cleared a long time ago and had been subdivided from farmland rather than native bush, although there are concerns for the remnants of native bush remaining if future subdivision occurs. In comments added to the survey, respondents identified that issues with pests and weeds could be coming externally, resulting from land use surrounding the valley, as well as land use from individual residents, and is an example of an issue requiring wider community cooperation to combat.

With over half of respondents indicating that important land uses on their property are using land for livestock and pets and that they are keen to achieve greater self-sufficiency in terms of food production, it is important to recognize that some negative effects and degradation of ecosystems services might result from land management practices that support these activities. For example, soil erosion and water pollution could result from livestock, or pets such as cats and dogs can cause predation on native birds. Of those respondents who said they had pets, just 10% said that they regularly kept their cats in at night to protect native wildlife. Although not all respondents specified what type of pets they owned, according to comments by some respondents, many residents have cats, which can cause serious issues for native birds if left to roam [52]. It was also noted that there is a problem with stray cats, and sometimes unwanted cats are left in the Valley from city residents. This could be having a detrimental effect on biodiversity [52] but would require more research to ascertain the impacts of cat ownership on biodiversity in the area.

Other threats to the area identified by respondents included water pollution and soil erosion (mostly from agriculture). Threats identified under "other" were largely external and were associated with development. In this category, respondents mentioned forestry operations, wind farm development, traffic and road issues, and housing development. Respondents also indicated that they wanted to protect a strong sense of community. They were particularly fearful of developmental threats and greater subdivision which would increase the intensity of housing and destroy the rural life that they love. Several respondents indicated they were against further subdivision and changes in minimum lot size that might threaten the space and privacy and peace, quiet, and tranquillity that living in the Valley currently provides. These views are consistent with submissions in the Palmerston North City Council's Rural-Residential Land Use Strategy [45], where most respondents were keen to maintain their rural lifestyle and therefore did not want to see a decrease in lot size. A wish to maintain the rural character of the Turitea Valley also provides opportunities to work with residents to maintain the character of the area.

Barriers that could potentially stop residents from engaging in environmental management practices predominantly included a lack of funds and/or time and, in some cases, a lack of knowledge and information. These factors should be considered by local council and conservation groups that operate in the area so that more targeted support and resources can be given to residents to help them to engage in looking after the ecosystem services on their property and within the wider peri-urban landscape, thus encouraging wider catchment environmental management by collective action. This support could include offering financial incentives to increase conservation efforts on these properties. These incentives could include reduced council rates for land "set aside" for conservation. They could also include a reduced price or free, native plants. Access to expert advice to assist with lifestyle block environmental planning could also help fill knowledge gaps. Capacity could also be built within the area through community initiatives and the creation of community groups that could assist each other with restoration work. Local catchment meetings/forums could be organized where residents and community advisers and/or the local council have discussions about the Turitea Valley and its future, which could also prove useful. Respondents identified online platforms as being a way that they are also happy to receive relevant information, so these could be utilized to provide regular newsletters, updates on social media, or via email.

Examples of workable community strategies can be drawn from programs established overseas that have helped support private landowners with the management of their properties to improve conservation and environmental outcomes. One such international environmental stewardship scheme is the Australian Land for Wildlife (LFW) program, which has been applied across most states and territories in Australia for the last 30 years [53]. The LFW program aims to improve wildlife habitat on private properties, and LFW officers provide personalized property plans, property maps, and any other technical information for the properties involved. In 2018, there were over 14,000 properties involved in the program covering 2.3 million ha with approximately 500,000 ha of habitat under voluntary protection and management [53]. The program is delivered through different agencies in different states and territories but mostly councils and has received some funding through the Australian Government's National Landcare Program. To date, LFW has had positive impacts on the environment by supporting landowners and providing advice for preserving the wildlife habitat on their properties for future generations. Some of the key topics LFW provide information on are creating habitat for specific species, restoring creeks, weed control, pest control, wildlife care, and rehabilitation. The program also provides information on wildlife corridors, wildlife-friendly fencing and netting, responsible ownership, the value of habitat trees, collecting and identifying plants, and developing weed management plans and weed control methods. The model this scheme provides and the practices it promotes could also be of significance when considering how to encourage activities in the Turitea Valley and other peri-urban environments across NZ to protect and enhance ecosystem services.

Whilst it is clear that much could be done in terms of biodiversity enhancement and protection in the peri-urban area by engaging residents, it is still a particular concern for NZ that lifestyle properties potentially reduce the amount of productive land [27]. Given its significance, most of the research undertaken on lifestyle properties in NZ has focused on the implications of this. Evaluating the consequences of taking high-value, versatile soils out of production for subdivision into lifestyle properties, thus impacting on the potential provisioning services that this previously important farmland contributed towards [27,54]. Loss of versatile soils has an impact on provisioning services at a national level, and this is not unique to NZ [55,56], but given that many "lifestylers" buy their properties with self-sufficiency in mind and not all of the land is built upon, this can mean that the property still maintains some provisioning services, but the scale is severely reduced. Food provisioning services on lifestyle blocks might only be available at a family or household scale. Food in this situation is no longer grown for domestic and international markets but instead grown for household consumption and maybe some local exchange or sale. Nevertheless, there

is potential on this land for lifestyle properties to contribute towards more local markets or join forces with the outputs from other properties to contribute smaller quantities of produce to a larger cooperative arrangement that could service a wider domestic market. With an increasing requirement for more sustainable farming practices in NZ, there is some value to be had from more locally grown produce. This means "lifestylers" could be encouraged to not just produce for themselves but also to produce to share. These small-scale production ventures would likely be low input and organic and provide outputs that could be sold at local markets, demonstrating even greater environmental benefits. With respondents to the survey listing climate change as their greatest environmental concern, being able to contribute towards a more localized market economy that has lower emissions might be worth their serious consideration. However, mobilizing such action would again require the creation of community groups that promote knowledge sharing and capacity building.

Peri-urban areas have been shown to make an important cultural service contribution to both peri-urban residents and urban ones [57]. This study also shows that lifestyle properties have an important role to play in contributing towards cultural services, especially at the household and community level, i.e., they provide for and support recreation, aesthetic and spiritual services for residents. However, they might also have a key role to play in providing some of these for urban residents of Palmerston North too. Their properties are visible in the urban–rural fringe, and the landscapes in which they reside are often places urban residents can visit on weekend trips out of the city. The Turitea Reserve, for example, is a place of recreation for Palmerston North, and therefore, the whole valley in which it sits plays an important role for the urban inhabitants of Palmerston North who want to experience local nature and nature-based activities. Peri-urban suburbs such as the Turitea also offer potential educational services to families who reside there with younger children who grow up with pets and stock and learn about the natural world in which they live.

The strong sense of cultural connection was evident amongst all respondents to the survey, and although the contribution of the peri-urban area from a cultural perspective is often not widely recognized, greater recognition must be given to the role that the landscapes play, especially when planning for future development is considered. In NZ, under the Treaty of Waitangi *Te Tiriti o Waitangi* (signed in 1840 between the British Crown and Māori chiefs), it is important to acknowledge the cultural importance of landscapes for Māori and to consider the views of local iwi and hapū (Māori tribes and clans) in modifications to the landscape [16]. Although this study did not consider Māori cultural connections to the Turitea landscapes, it is important that Māori beliefs, values, and connections are considered in any development plans. Creating community-led ecosystem service protection activity could include not just "new" residents but also traditional custodians of the land.

This study also shows that lifestyle properties have a potential role to play in the provision of regulating services, e.g., in helping with water quality control, soil erosion, and pest regulation. Land management practices such as native planting and weed and pest control can help in the provision of regulatory services [58–60]. This study has shown that more can be done in terms of weed and pest control in this area. This could be useful for urban environments as well as rural ones that lie beyond the peri-urban area. Lifestyle properties also offer a potential role in pollination, with some beekeeping being noted on blocks in the study, but given the extensive garden focus by residents, increasing native and other flowerings species could also help with pollination in the area [61]. Extra tree planting can also serve to act as carbon storage and assist in flood control, whilst additional planting can help to stabilize highly erodible areas [60].

Given the lack of expectation of deriving an income from their land compared to farmers, this demonstrates that lifestyle property residents in the peri-urban area should be considered separate from farmers, and the drivers and motivators behind land management will be very different, so current approaches targeted at improving "on farm" sustainability might not be as effective or relevant to "lifestylers". They also potentially have different

values that they place on the land than urban residents, which provides good opportunities for ecosystem service management in this important area on the urban fringe.

In summary, the threat to the market economy of NZ through the expansion of the peri-urban fringe to incorporate more lifestyle properties through loss of versatile soils and land that could have grown export crops has been acknowledged [27,54]. The potential environmental threats that can arise if "lifestylers" do not manage their properties well is also a risk. However, there are important benefits to be had if consideration is given to the potential for these properties to be managed for biodiversity and other supporting and regulating ecosystems services. There is a population that lives in these areas that appears keen to preserve a particular way of life, to maintain peace and tranquillity, to encourage native flora and fauna, and to discourage exotic and introduced pest species. They offer opportunities for lower intensity stock management and can assist in erosion-prone areas by planting appropriate species that can reduce sediment loss and stabilize hill slopes. They can also assist in water quality control by planting riparian strips, recreating wetlands, and protecting water systems. With the right encouragement towards strategic native plantings, they can contribute to reducing fragmentation of habitat whilst also assisting in the ongoing monitoring of native species and ecosystem and landscape health. These are all land management activities that are being increasingly called for to more sustainably manage NZ's landscapes so as to be able to live up to NZ's "clean green" reputation. This survey shows that with the right motivation and incentives, it might be possible to engage this population in collective community lead ecosystem service protection which can help to look after the landscape ecology of peri-urban areas, but it will also need planning controls that acknowledge the important values and services that the peri-urban environment can provide [62].

This study serves to provide a good indication of what lifestyle block residents in the Turitea Valley think and feel about land management on their properties and what role they could play in ecosystem service management and protection. It provides a useful insight to lifestyle property residents that has not been investigated previously. However, some limitations need to be acknowledged. The response rate was deemed reasonable, but why people did not take part should be considered and explored further to improve study validity [63]. With non-response bias presenting a likely limitation, the results should be viewed with caution and interpreted within the context of exploratory findings that could form the basis of further investigation.

The demographics of survey respondents are different from the 2013 census data for Turitea. Although the survey is conducted on a subset of the Turitea, i.e., the Turitea Valley and lifestyle properties of a particular size, the results suggest that not all demographics might be represented. Responses to the survey were also heavily skewed towards people over 50 years of age and with a Bachelor's degree or higher—the higher than the average level of education could be attributed to the fact that this particular suburb is close to Massey University and is a popular place to live for staff and students. The block size of respondents who engaged in the survey was also at the smaller end of the scale, which also could influence land-use practices and responses to particular questions around land management. It is also likely, given that the survey was voluntary, that there is some response bias towards people who are more concerned about environmental issues and are pro-environmental.

## 5. Conclusions

Within NZ, as is the case in many countries, there is increasing pressure for more houses to accommodate a growing population. Urban developments and subdivisions in the peri-urban areas are a response to the growing demand for houses. In association with this, the number of lifestyle properties is increasing. People wanting to escape urban areas and move into quieter and more rural spaces are creating demand for accommodation to satisfy a "tree change" [28]. This "tree change" phenomenon is possibly likely to increase with recent COVID-19 restrictions and lockdowns and the increased move towards people

working from home [64]. As people are required to spend less time in the office, more people may choose to move out of urban areas and chase a more rural way of living whilst engaging in telework. Those making this move are generally not moving for financial purposes, such as generating an income from their lifestyle property but are, as the name suggests, making lifestyle choices. However, as this subdivision occurs, it swallows up agricultural land reducing the potential of that land from a provisioning service capacity, but with increasing environmental challenges facing both urban and rural agricultural landscapes, lifestyle blocks in the peri-urban have an opportunity to play an important role in protecting natural capital and restoring important ecosystem services. They have the potential to put back native vegetation that was cleared to create the farmland that was then utilized in their subdivision. They also have the potential to contribute at a small scale to local food production.

This exploratory study indicates that lifestyle property residents potentially have a good sense of environmental stewardship and a desire to plant native species and protect their land from the invasion of pests and weeds. These residents are also often quite community-focused and protective of their special place. This creates an excellent basis from which to encourage greater action towards protecting and enhancing biodiversity and to put in place other land management strategies that can enhance natural capital and assist in other ecosystem service protection. To do this, there needs to be a focus on support and provision of the environmental management resources required by those residing on these properties to make land management changes. They need to have all the information required on maintaining their block of land while protecting the environment at the same time and preserving wildlife habitats for future generations.

Properties such as those found in the Turitea Valley have the potential to be managed for the protection and conservation of important natural capital and ecosystem services if the community work alongside the council and government to create and maintain practical and realistic environmental management goals. This survey points to the creation of support services and platforms in the form of online information, advisors, and local group meetings. It also shows that there is an opportunity to build knowledge and capacity through community initiatives and groups. With respondents indicating they feel a strong connection to the Turitea Valley, developing a platform for community interaction and cooperation towards positive environmental outcomes for the area may enhance this feeling of connection and translate it towards greater ecosystem service provision within the Turitea Valley. Implementing these measures to specifically focus on local communities such as the Turitea Valley and their needs will be more effective in engaging residents because issues are local and relevant. There is also potential to offer incentives to increase conservation efforts on these properties. These incentives could include reduced rates for land set aside for conservation or reduced price or free native plants and access to expert advice to assist with lifestyle block environmental planning. Supporting this could be the creation of environmental policy that guides land use on lifestyle developments towards management for wider community and catchment benefits, especially promoting management for each category of ecosystem services (supporting, regulating, provisioning, and cultural). However, "bottom up" initiatives are likely to be more palatable than any regulatory ones and have the most beneficial outcomes.

For countries like NZ that face large challenges around water quality and erosion associated with an intensive agricultural industry, lifestyle properties offer great potential to assist in the recreation or protection of important landscape structures that can assist in the preservation of important landscape processes or help to reduce detrimental ones that result in land degradation and water pollution. To help the landscape ecology of the peri-urban area, lifestyle property residents could form a "green army" that puts an important ecological buffer around urban areas and connects it to important remnants in the wider catchment. This could help to create a sanctuary for native species on the border of agricultural areas. So as farmers also respond to increasing pressure to restore biodiversity and manage their land more sustainably, the extra potential is created to provide important

connectivity and reduce habitat fragmentation across the wider landscape. This means that peri-urban landscapes and the lifestyle block could have an increasingly important role to play in ecosystem service provision and management in NZ (and overseas) at the catchment scale. However, it will need community encouragement to show the residents just what an important role they can play, with appropriate incentives and an increased focus on building knowledge capacity to help them create the right patchwork of habitat and pest control for the best landscape ecological outcomes.

**Funding:** This research received no external funding but was supported under the Massey University Research Fund 2018. Palmerston North City Council, in association with Massey University's Living Lab Project, also provided support for two summer scholarships for students to intern with the School of Agriculture and Environment to work on the project.

**Data Availability Statement:** Data are available on request due to restrictions on privacy and ethical grounds. The data are not publicly available to ensure confidentiality.

**Acknowledgments:** Acknowledgement is given to Massey University and Palmerston North City Council for supporting this research. I would also like to thank Alannah Hoskin and Lisa Boterman for their hard work on the project as part of their summer internship. I would also like to thank the valuable advice given by 3 anonymous reviewers that helped to improve the manuscript and Julian Gorman and Richard Aspinall for their comments on earlier versions of the paper.

**Conflicts of Interest:** The author declares no conflict of interest.

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
