# Peer review of "Lifestyle Properties, Ecosystem Services, and Biodiversity Protection in Peri-Urban Aotearoa–New Zealand: A Case Study from Peri-Urban Palmerston North"

_land, doi:10.3390/land10121345_

Round 1

Reviewer 1 Report

The paper uses a qualitative study method investigated the growing lifestyle land development in the peri-urban Palmerston North area. The study background, methodology and results are well presented. The discussion is thorough and insightful. It is recommended to publish after the following revisions:

  • please do a thorough language and spelling check before publication, some of the minor errors need to be corrected, e.g. line 154 "th planting"
  • it is suggested to redo the study area map in Figure 1 by using GIS software or reference the study area by local official maps. Showing the study area in a clear way will help readers to understand better of the study topic. 

Author Response

Thank you for reviewing my paper and suggesting useful improvements.  

I have completed a thorough spell-check and language review of the paper and updated it accordingly.

I have also redone figure 1 as suggested.

Reviewer 2 Report

I really have doubts about this:

A total of 32 responses that are enough to draw conclusions...

Author Response

Thank you for reviewing my paper.

I have taken on board your concerns about the number of respondents.   I have adjusted the wording in places to enforce that the study is only exploratory and that it just gives an indication, recognizing that for more concrete conclusions to be drawn more in-depth studies would be required.   And I have added some information on the implications of the lower response rate.  

Reviewer 3 Report

Title is very wordy -- could it be: "Life-style properties, ecosystem services and biodiversity conservation in peri-urban Aotearoa-New Zealand" ?

Figure 1 -- please improve the contrast in the map

Please frame sharper questions/hypotheses as these can help to get more focus in the discussion and conclusion sections which are rather long and could benefit from more focus

Author Response

Thank you for taking the time to review my paper and for making some useful suggested recommendations to improve it.

I have shortened the title as suggested.  

I have redone figure 1 so the contrast is much brighter.

I have more explicitly stated the aims of the study and the research question (see lines 143 to 148 and 156 to 163) and made sure that the discussion and conclusion sections link to this.